# Bloodstream Infections by AmpC-Producing Enterobacterales: Risk Factors and Therapeutic Outcome

**DOI:** 10.3390/pathogens12091125

**Published:** 2023-09-03

**Authors:** Mladen Pospišil, Haris Car, Vesna Elveđi-Gašparović, Nataša Beader, Zoran Herljević, Branka Bedenić

**Affiliations:** 1Krapina-Zagorje County Community Health Centre, 49245 Stubica, Croatia; mladenpospisil@gmail.com; 2Zagreb Health School, 10000 Zagreb, Croatia; carharis112@gmail.com; 3Department of Gynecology and Obstetrics, School of Medicine, University of Zagreb, 10000 Zagreb, Croatia; vesnagasparo@gmail.com; 4Department of Gynecology and Obstetrics, University Hospital Centre Zagreb, 10000 Zagreb, Croatia; 5BIMIS—Biomedical Research Center Šalata, School of Medicine, University of Zagreb, 10000 Zagreb, Croatia; natasaeli@gmail.com; 6Clinical Department for Clinical and Molecular Microbiology, University Hospital Centre Zagreb, 10000 Zagreb, Croatia; zoranhe@gmail.com

**Keywords:** antimicrobial resistance, AmpC-producing Enterobacterales, bloodstream infections

## Abstract

Bloodstream infections associated with AmpC-producing Enterobacterales are severe medical conditions which, without prompt and effective treatment, may have dire ramifications. This study aimed to assess whether certain comorbidities and previous surgical procedures coincide with resistance determinants of AmpC-producing Enterobacterales associated with bloodstream infections. Antibiotic resistance patterns and therapy outcome were also determined. The patients’ data obtained revealed that the prevalence of recent surgical procedures, solid organ tumors, metabolic diseases, kidney and liver failure, and hematological malignancies do not differ between resistant and susceptible isolates of AmpC-producing Enterobacterales. Furthermore, no difference was reported in mortality rates. Regarding antibiotic resistance, 34.52% of isolates were confirmed to be resistant (AmpC hyperproduction, ESBL, or carbapenemase). More than one in five AmpC hyperproducers were reported amid *Providencia* spp., *K. aerogenes*, *E. cloacae*, and *C. freundii.* strains. Carbapenemases were mostly noted in *Providencia* spp. followed by *M. morganii* and *K. aerogenes* strains. *Serratia marcescens* had the highest proportion of ESBLsof ESBLs. Resistance to expanded-spectrum cephalosporins of *Providencia* spp. and *K. aerogenes* strains exceeded 50%, and resistance to meropenem over 10% was observed only in *C. freundii* strains. Enterobacterales’ ever-growing resistance to antibiotics is becoming quite a challenge for clinicians and new treatment options are required.

## 1. Introduction

Bloodstream infections (BSIs) are severe medical conditions and one of the leading causes of morbidity and mortality around the world. Hospital-acquired bloodstream infections (HA-BSIs) are healthcare-associated infections that develop 48 h or more after hospital admission and are commonly linked to intensive care units (ICUs). By comparison, community-acquired bloodstream infections (CA-BSIs) are infections that become clinically evident within 48 h of hospital admission [1].

Increased prevalence of multidrug resistant (MDR) pathogens, both in community and hospital surroundings, combined with inappropriate antimicrobial therapy, is a major contributing factor to prolonged illness and lethal outcome [2]. Treatment of invasive infections caused by MDR Gram-negative bacteria has proven to be a challenging endeavor since advanced antimicrobial medications are not available in some countries. Such deficiency of potent antibiotics hinders clinicians’ efforts in combatting septicemia.

Treatment failures are usually associated with methicillin-resistant *Staphylococcus aureus* (MRSA), vancomycin-resistant *Enterococcus faecium* (VRE), extended-spectrum β-lactamase (ESBL), AmpC-producing Enterobacterales, carbapenem-resistant Enterobacterales (CRE), carbapenem-resistant *Pseudomonas aeruginosa* (CRPA), and *Acinetobacter baumannii* (CRAB) [3]. They belong to the so-called ESKAPE pathogens [4,5]. Alleged contributable risk factors for acquiring bloodstream infections caused by multidrug resistant pathogens include frequent administration of antibiotics, indwelling urinary and intravascular catheters, recent surgical procedures, and prolonged stay in a hospital intensive care unit.

AmpC β-lactamases are primarily cephalosporinases encoded by chromosomes or plasmids (pAmpC). Chromosomal AmpC β-lactamases are produced by *Enterobacter cloacae*, *Klebsiella aerogenes, Providencia* spp. *Serratia marcescens*, *Citrobacter freundii*, and *Morganella morganii* [6]. Their hydrolytic activity involves penicillins, first-, second-, and third-generation cephalosporins, and cephamycins sparing cefepim and carbapenems. Unlike ESBLs, inhibition does not occur via the activity of clavulanic acid, sulbactam, or tazobactam; however, some β-lactam combinations with inhibitors such as piperacillin/tazobactam may prove effective against AmpC-positive organisms [7].

Plasmid-mediated AmpC β-lactamases (pAmpC) are derived from the chromosomal β-lactamases of the bacteria belonging to the genii *Enterobacter*, *Serratia*, *Citrobacter*, *Pseudomonas*, and *Acinetobacter* due to a transfer of the chromosomal gene to the plasmid [6,7]. The most prevalent types are DHA, ACT, FOX, MOX, MIR, LAT, ACC, and CMY, usually detected in *K. pneumoniae*, *E. coli*, and *P. mirabilis*. *bla*_AmpC_ genes are found adjacent to an insertion sequence common region (ISCR1) involved in gene mobilization into, typically, complex class 1 integrons. Organisms producing high quantities of AmpC β-lactamase typically produce a positive ESBL screening test, but test negative in an inhibitor-based test with clavulanic acid. They are confirmed by inhibitor-based tests with phenylboronic acid or cloxacillin [7]. Exposure to β-lactam antibiotics can induce a high level of AmpC expression, leading to resistance to some β-lactam antibiotics, most notably expanded-spectrum cephalosporins (ESCs) [6,8]. In addition, mutations of the genes that regulate AmpC expression may result in derepression of AmpC β-lactamases, causing a selection of resistant mutants during antibiotic therapy [8]. Enterobacterales with chromosomal AmpC β-lactamases are prone to acquiring additional resistance traits such as ESBL or carbapenemases.

ESBLs are plasmid encoded β-lactamases capable of conferring bacterial resistance to penicillins, first-, second-, and third-generation cephalosporins, and aztreonam [9,10]. They are predominantly found in *Klebsiella pneumoniae* and *Escherichia coli,* but can be present as additional β-lactamases in AmpC-producing Enterobacterales [9].

Carbapenemases associated with Enterobacterales belong to Ambler class A serine β-lactamases (KPC, GES, SME, IMI, NMC), class B metallo-β-lactamases (MBL) of the IMP, VIM, or NDM family, or OXA-48-like β-lactamases belonging to the class D or carbapenem-hydrolyzing oxacillinases [11,12]. 

Characteristics of bloodstream infections caused by AmpC-producing Enterobacterales have not been extensively investigated and only few reports in the medical bibliography exist [13,14]. The previous reports on AmpC β-lactamases from Croatia were related to p-AmpC in *P. mirabilis* [15] and *K. pneumoniae* [16]. The effects of hyperproduction and induction of AmpC β-lactamases on the susceptibility to β-lactam antibiotics in *E. cloacae* were also investigated [17]. However, those studies were only in vitro investigations without clinical correlations. Recognizing causative agents of BSIs by AmpC-producing Enterobacterales, their resistance determinants, and their antibiotic resistance patterns, plays a pivotal role in the prompt and beneficial administration of appropriate antimicrobial medications [2] which differentiates between life and death. 

The reports on the BSIs associated with AmpC-producing Enterobacterales are scarce in the medical bibliography and are mostly focused on p-AmpC β-lactamase producing *E. coli* and *K. pneumoniae* or therapeutic outcome; therefore, adequate research was deemed necessary.

This study aimed to assess whether certain comorbidities and previous surgical procedures coincide with resistance determinants of AmpC-producing Enterobacterales associated with bloodstream infections in University Hospital Center Zagreb (UHCZ). Antibiotic resistance patterns, resistance determinants, and therapeutic outcome were also determined. 

## 2. Materials and Methods

### 2.1. Patients 

A study was conducted involving 195 patients with bloodstream infection caused by isolates positive for chromosomal AmpC β-lactamase: *E. cloacae*, *K. aerogenes*, *Serratia marcescens*, *Morganella morganii, Providencia* spp., *Citrobacter freundii*, and *Enterobacter* spp. Of 195 patients, supporting clinical data were available for 172 patients (107 males and 65 females, age range 0–88, median 57.5); therefore, 23 patients who were hospitalized in the children’s hospital near Zagreb, were excluded. Their blood cultures were analyzed, nevertheless. Twenty preterm newborns were also subjects of this study. CA-BSI was defined as a positive blood culture (BC) sample acquired within 48 h of hospital admission, while HA-BSI was defined as a positive blood culture acquired at least 48 h after hospital admission. A total of 172 patients were hospitalized in different hospital wards of the University Hospital Centre Zagreb (UHCZ), including the medical ward, surgery, hematology, pediatrics, gynecology and obstetrics, and intensive care units from the 1st of January, 2020 until the 31st of December, 2022. Cases were detected by obtaining blood culture results.

A bloodstream infection (BSI) was diagnosed based on the presence of causative agents in the blood accompanied by systemic inflammatory response (elevated or decreased temperature (>38.5 °C or <36.0 °C), increased heart rate (>90), respiratory rate (>20) and leukocytosis (≥12,000/mm^3^), or leukopenia (≤4000/mm^3^) [1]. Severe sepsis was defined as sepsis complicated by organ dysfunction with reversible hypotension after volume replacement therapy and septic shock was defined as a sepsis complicated by organ dysfunction and persistent hypotension despite volume replacement therapy [18]. Two blood samples were taken under aseptic conditions from clinically diagnosed septicemia episodes for routine blood culture before initiating antibiotic therapy. Blood culture (BC) bottles were used for routine cultivation and diagnosis, which included BACTEC FX (Becton-Dickinson, New Jersey, USA). Positive BCs were subjected to Gram staining and subcultured on solid medium (blood agar, chocolate agar, and Columbia anaerobic blood agar), and after 18 to 24 h incubation (overnight), were identified by a MALDI-TOF MS (matrix-assisted laser desorption ionization-time of flight mass spectrometry) Biotyper (Bruker, Daltonik GmbH, Bremen, Germany) [19].

Patients’ demographic features and attributed risk factors were obtained by reviewing the hospital charts. The following data were collected from the hospital information system (LIS): age, gender, diagnosis, comorbidities, source of BSI, surgical procedure, antibiotic therapy, and outcome. In order to determine whether prevalence of metabolic diseases, solid organ tumors, hematologic malignancies, kidney and liver failures, and recent surgical procedures differ between susceptible and resistant strains, we used the chi-square test of independence. The chi-square test of independence is a statistical test used to detect the presence of association between two categorical variables. The test involves the creation of a contingency table that cross-tabulates the frequencies of the two variables’ categories. The chi-square test statistic is calculated by comparing the observed and expected frequencies from the contingency table, whereas the associated *p*-value is found by comparing the test statistic to a chi-square distribution with degrees of freedom determined by the table dimensions. A *p*-value < 0.05 was considered statistically significant. 

The study was approved by the Ethical Committee of the UHCZ (number 02/21 AG, class 8.1-23/66-2). 

### 2.2. Bacterial Isolates

The Enterobacterales with chromosomal AmpC β-lactamases: *Enterobacter cloacae*, *K. aerogenes*, *Serratia marcescens*, *Morganella morganii, Providencia* spp., *Citrobacter freundii*, and *Enterobacter* spp. were isolated from blood cultures between 1 January 2020 and 31 December 2022.

### 2.3. Antibiotic Susceptibility

The antibiotic susceptibility data were obtained from the hospital information system (LIS). The disk diffusion test was performed for the purpose of routine microbiology diagnostics, according to EUCAST guidelines [20], except for colistin, for which the broth dilution method was used. The panel for antibiotic susceptibility testing included the following antibiotics: piperacillin/tazobactam, ceftazidime, cefotaxime, ceftriaxone, cefepime, ertapenem, imipenem, meropenem, gentamicin, amikacin, and ciprofloxacin. Additionally, colistin and ceftazidime/avibactam were tested for carbapenemase-producing organisms. ESBLs were screened based on reduced inhibition zones around cephalosporin disks and confirmed by a double disk synergy test [21,22]. *E. coli* ATCC 25922 and *K. pneumoniae* 700603 were used as negative and positive control strains. Overproduction of AmpC β-lactamases was inferred based on resistance to third-generation cephalosporins and cefoxitin and while preserving susceptibility to cefepime and carbapenems. Production of carbapenemases was suspected based on reduced inhibition zones around carbapenem disks according to EUCAST criteria (less than 22 mm for imipenem and meropenem, and less than 25 mm for ertapenem) and confirmed by the immunochromatographic OKNV (OXA-48, KPC, NDM, VIM) test [23].

## 3. Results

### 3.1. Patients

Of 172 patients supported by corresponding clinical data, 54 (31.40%) patients were hospitalized in medical wards, 45 (26.16%) in the emergency unit, 36 (20.93%) in each of the surgical and ICU wards, and 1 (0.58%) in a COVID-19 unit (Figure 1).

Patients’ bloodstream infection episodes were divided into hospital (73.84%) or community (26.16%) acquisition. Forty-one patients (23.84%) had a recent surgical procedure. Solid organ tumors were detected in 31 (18.02%) patients, metabolic diseases in 33 (19.19%), kidney and liver failure in 46 (26.74%), and hematological malignancies in 12 (6.98%).

Fifty-four patients out of 172 (31.40%) had resistant and 118 (68.60%) susceptible strains. Using the chi-square test of independence, no statistically significant difference was detected when comparing resistant and susceptible strains in terms of prevalence of metabolic diseases (24.07% vs. 16.95%, *p* = 0.27) solid organ tumors (16.67% vs. 18.64%, *p* = 0.75), hematologic malignancies (7.41% vs. 6.78%, *p* = 0.88), kidney and liver failures (31.48% vs. 24.57%, *p* = 0.34), and previous surgical procedures (20.37% vs. 25.42%, *p* = 0.47).

In HA-BSIs, the proportion of cases with resistant strains was higher than that in CA-BSIs, but again the difference was not statistically significant (32.2% (41/127) vs. 28.89% (13/45), *p* = 0.67). 

Regarding the source of BSI, it originated from the abdomen in 27 (15.70%) and from the urinary tract in 26 (15.12%) patients. CVC and other catheters were the source of BSI in 24 (13.95%) patients. Association with febrile neutropenia was noted in 18 (10.47%) patients and with pneumonia in 16 (9.30%) patients. The source of BSI was identified as umbilical cord in eight (4.65%) patients, as skin and soft tissues in four (2.33%) patients, and chorioamnionitis in three (1.74%) patients. The BSI source was deemed cryptogenic in 46 (26.74%) patients (Figure 2).

### 3.2. Bacterial Isolates

The study comprised 197 isolates: 87 *E. cloacae*, 16 *K. aerogenes*, 47 *Serratia marcescens,* 9 *Morganella morganii,* 11 *Providencia* spp., 9 *Citrobacter freundii*, and 18 *Enterobacter* spp. strains, as shown in Table 1. Of 197 isolates, 129 isolates were confirmed to be susceptible (65.48%) and 68 (34.52%) resistant (AmpC hyperproduction, ESBL, or carbapenemase). The number of bacterial isolates was increased by two, in comparison with the number of patients, due to the fact that two different AmpC-positive strains were identified in two different patients’ blood cultures. The proportion of blood cultures with AmpC-producing Enterobacterales was 197/39,098 (0.50%), where the denominator is the total number of BCs undertaken in a certain timeframe in UHCZ.

### 3.3. Antibiotic Resistance Patterns and Determinants 

*E. cloacae* showed the highest resistance rates to ESC (cefotaxime 35.63%, ceftazidime and ceftriaxone 34.48%), followed by cefepime (27.59%), piperacillin/tazobactam (24.14%), ciprofloxacin and gentamicin (20.69%), as shown in Table 1. Meropenem and imipenem maintained good activity with only 5.75% of resistant isolates, respectively, while amikacin was efficient in 98.85% of isolates. Resistance to colistin was not reported. Twenty isolates (22.99%) were Amp-C hyperproducers, whereas ESBLs and carbapenemases were detected in ten (11.49%) and four isolates (4.60%), respectively. OKNV identified only MBLs (two VIM and two NDM) (Table 1). One isolate exhibited resistance to ertapenem but tested negative for carbapenemases.

*K. aerogenes* demonstrated higher resistance rates to ESC (62.50%) and piperacillin/tazobactam (62.50%). Cefepime, gentamicin, and ciprofloxacin exhibited moderate activity with 31.25%, 25.00%, and 18.75% of resistant isolates, respectively, as shown in Table 1. Amikacin and carbapenems preserved activity on the majority of isolates with 12.50% and 6.25% of resistant isolates, respectively. Overproduction of Amp-C and ESBLs was identified in three isolates (18.75%), respectively. OXA-48 was found in two carbapenem resistant isolates (12.50%). 

Resistance rates of *Serratia marcescens* to ESC were as follows: ceftazidime, 27.66%; ceftriaxon, 27.66%; and cefotaxime, 25.53%. Resistance to cefepim and gentamicin was noted in 23.40% and 25.53%, respectively. Amikacin, piperacillin/tazobactam, ciprofloxacin, and carbapenems were active against all except 14.89%, 8.51%, 8.51%, and 4.25% of the isolates. High rates of ESBL production (19.15%) and AmpC hyperproduction (17.02%) were reported. OXA-48 was detected by OKNV in only one isolate, whereas the other carbapenem-resistant strains tested negative for carbapenemase production.

A high efficacy of antibiotics was demonstrated against *Morganella morganii*, with a 22.22% resistance rate to cefepime and piperacillin/tazobactam. ESC showed activity against 88.89% of the isolates, as shown in Table 1. Rates of resistance to ciprofloxacin, carbapenems, and aminoglycosides were not recorded. ESBLs and Amp-C hyperproducers were not identified, but two isolates produced OXA-48, which conferred only ertapenem resistance on the producing isolates, whereas imipenem and meropenem remained susceptible.

*Providencia* spp. displayed over 50% resistance rates to the majority of antibiotics used in the treatment of BSIs, including ESC (63.64%), gentamicin (63.64%), and ciprofloxacin (72.73%). Regarding carbapenems, a resistance rate of 27.27% to imipenem was noted while strains remained susceptible to meropenem. Five out of 11 isolates (45.45%) hyperproduced AmpC: three (27.27%) were carbapenemase OXA-48 producers and two (18.18%) tested positive for an ESBL (Table 1).

Regarding *Citrobacter freundii*, moderate resistance rates of 33.33% were reported for ESC, piperacillin/tazobactam, gentamicin, and ciprofloxacin, as shown in Table 1. Hyperproduction of AmpC β-lactamase was identified in two isolates. Only one isolate tested positive for an ESBL and carbapenemases, respectively. One carbapenemase-positive strain harbored two carbapenemases: KPC and NDM.

*Enterobacter* spp. showed high susceptibility (over 80%) to all antibiotics, as shown in Table 1, with only one strain being positive for an ESBL. AmpC hyperproduction was noticed in three isolates.

### 3.4. Antibiotic Therapy Outcome

A lethal outcome was reported for 37 out of 172 patients (21.51%), of which 27 (72.97%) died from BSI due to septic shock (attributable mortality) and the rest due to progression of underlying disease (all-cause mortality). The mortality rate of CA-BSI of 33.33% (15/45) was reported, whereas lethal outcome occurred in 17.32% (22/127) of the patients with HA-BSI. Mortality rates associated with individual bacterial species were as follows: *E. cloacae* (17.24%), *K. aerogenes* (18.75%), *S. marcescens* (25.53%), *M. morganii* (33.33%), *Providencia* spp. (9.09%), *C. freundii* (11.11%), and *Enterobacter* spp. (11.11%). Bloodstream infections caused by resistant strains resulted in a higher mortality rate compared to BSI with a susceptible strain (24.07% vs. 20.34%) but the difference was not significant (*p* = 0.58). Clinical cure at the end of therapy was achieved in 78.52% of the patients and complete eradication of the pathogen was noticed in 147 (93.04%) surviving patients.

According to the clinical data, as an empirical therapy, the most frequently administered antibiotic was meropenem, with an application rate of 13.95% (24 patients), followed by a combination of meropenem and vancomycin, with an application rate of 8.72% (15 patients). Subsequent to antibiotic susceptibility testing, the antibiotic with the highest application rate was meropenem, which was used in over a quarter of the total number of patients (29.07%, 50 patients), followed by a combination of meropenem and vancomycin (15.12%, 26 patients) (Figure 3).

Prior to antibiotic susceptibility testing, application rates of ceftriaxone and cefepime of 4.07% (seven patients) and 1.16% (two patients), respectively, were observed. Following the susceptibility test, the application rate of ceftriaxone decreased (2.33%, four patients), while cefepime’s increased (2.91%, five patients) (Figure 3).

The observation was made that piperacillin/tazobactam was used in eleven patients (6.40%) as an empirical therapy, and in fifteen patients (8.72%) as a targeted therapy, following antibiotic susceptibility testing (Figure 3).

As a result of ciprofloxacin’s broad-spectrum efficacy, application rates of 7.56% (13 patients) prior to antibiogram and of 11.05% (19 patients) after antibiotic susceptibility testing were observed (Figure 3). 

Use of a combination of gentamicin and other antibiotics was observed in twelve patients (6.98%) prior to antibiotic susceptibility testing; however the application rate decreased to 1.74% (three patients) after the results (Figure 3).

Application rates of amikacin as an empirical therapy and as a targeted therapy of 1.16% (2 patients) and 6.40% (11 patients), respectively, were noted (Figure 3). 

As an empirical therapy, the administration of the last-line antibiotic colistin was reported in seven patients (4.07%) colonized with *Acinetobacter baumannii,* whereas it was used in nine patients (5.23%) with carbapenem-resistant strains subsequent to antibiotic susceptibility testing (Figure 3).

Administration of ceftazidime/avibactam to four patients (2.33%) was reported following identification of OXA-48 carbapenemase (Figure 3).

Antibiotic switch was observed in 61 patients (35.47%) subsequent to antibiotic susceptibility testing.

Sporadic failures of microbiological eradication were noticed while administering penicillins, meropenem, and ciprofloxacin. It is estimated that AmpC-positive strains were eradicated using meropenem.

## 4. Discussion

The main finding of the study was that, contrary to prevalent opinions, prevalence of recent surgical procedures, solid organ tumors, metabolic diseases, kidney and liver failure, and hematological malignancies do not differ between resistant and susceptible isolates of AmpC-producing Enterobacterales. In this study, the majority of BSI episodes were cryptogenic, while the primary known source in the adults was the abdomen. Indwelling vascular and umbilical catheters were the dominant source of bloodstream infection in preterm newborns. This is in contrast with a previously published study which found the urinary tract to be the dominant source of ESBL-producing *E. coli* as causative agents of BSI [24]. CTX-M β-lactamases were the most prevalent in their study, whereas molecular characterization of ESBLs was not performed in this study. Equal rates of HA-BSI and CA-BSI were noted in previous studies [24]; however, in our study, hospital infections were significantly more represented, which is in concordance with the prevalence in Brazil (70.09%) [25]. 

*E. cloacae* was the dominant species associated with BSI, which is similar to the finding of a Canadian study [13]. In a Nepalese study, remarkably higher resistance rates were observed in *E. cloacae* compared to our results, for cefotaxime (100% vs. 35.63%), gentamicin (82.28% vs. 20.69%), and ciprofloxacin (60.76% vs. 20.69%) [26]. The resistance rates to imipenem and meropenem were similar (7.59% vs. 5.75% and 5.06% vs. 5.75%, respectively). On the contrary, the Italian study [27] reported lower resistance rate to ESC (56% vs. 62.50%). Moreover, rates of resistance to ciprofloxacin, gentamicin, and imipenem were also lower (18% vs. 20.69%, 12% vs. 20.69%, and 1% vs. 5.75%, respectively). On the other hand, in our study, a lower rate of resistance to amikacin compared to the Italian study was noted (1.15% vs. 10%) [27]. 

Concerning *K. aerogenes,* the Italian study reported higher resistance rates to amikacin and ciprofloxacin (37% vs. 12.50% and 30% vs. 18.75%, respectively), whereas resistance rates to ceftazidime (56% vs. 62.50%), cefotaxime (41% vs. 62.50%), gentamicin (6% vs. 25.00%), and imipenem (1% vs. 6.25%) were lower in comparison to our study [27].

Resistance rates of *Serratia marcescens* to tested antibiotics did not exceed 30%, which coincides with both Iranian [28] and Nigerian research [29]. In the Nigerian study, compared to ours, resistance rates to ceftriaxone were 22.78% vs. 27.66%, to ceftazidime were 19.62% vs. 27.66%, and to cefotaxime were 0.63% vs. 25.53%, while in the Iranian study, compared to ours, the resistance rates to the aforementioned antibiotics were 26.67% vs. 27.66%, 20.00% vs. 27.66%, and 23.33% vs. 25.53, respectively. In the Iranian [28] study, resistance to carbapenems was non-existent, yet in the Nigerian [29] study, 12.66% of strains were resistant to meropenem and imipenem, which was a higher resistance rate than the one observed in our study (4.25%). In the Nigerian study, the resistance rate to ciprofloxacin was 4.43%, which was almost half that in our study (8.51%). Resistance to ciprofloxacin was not reported in the Iranian study. Piperacillin/tazobactam was applied in all three studies. Resistance rates in the Nigerian, Iranian, and present studies were 19.62%, 10.00%, and 8.51%, respectively [28,29].

Regarding *Morganella morganii,* an Australian [30] study found resistance rates to fluoroquinolones, aminoglycosides, and carbapenems of 9.45%, whereas no resistance to these antibiotics was recorded in our study. On the one hand, the resistance rates of *M. morganii* to ceftazidime were higher in the Australian study compared to ours (18.10% vs. 11.11%). On the other hand, in our study, lower resistance rates to ceftriaxone, cefepime, and piperacillin/tazobactam were recorded (11.11% vs. 8.29%; 22.22% vs. 1.80%; and 22.22% vs. 3.88%, respectively) [30].

Resistant rates of *Providencia* spp. were extraordinarily high, but nonetheless coincides with Chinese research [31]. In the Chinese study, resistance rates to ceftazidime, cefotaxime, and ceftriaxone were 39.47%, 55.26%, 64.47%, respectively, while in our study only 36.36% of the strains were susceptible to ESC. Equal prevalence of resistance was found to cefepime (36.84% vs. 36.36%). Resistance to carbapenems was low in both studies, with only 32.89% and 27.27% of strains showing imipenem resistance in the Chinese and present studies, respectively. Resistance rates to gentamicin and ciprofloxacin were higher in our study compared to the Chinese study (63.64% vs. 44.74% and 72.73% vs. 38.19%, respectively). Amikacin proved susceptible to 72.72% of the strains in our study and to 61.84% of the strains in the Chinese study [31].

In this study, as many as one in three strains of *Citrobacter freundii* were resistant to the used antibiotics. A similarly high susceptibility rate was also found in a Taiwanese study [32]. They report resistance rates of 2.44% to gentamicin, 7.32% to ciprofloxacin, and 10.98% to amikacin. Regarding ESC, in our study, a resistance rate of 33.33% was observed, while in the Taiwanese study, the proportion of strains resistant to ceftazidime, cefotaxime, and ceftriaxone was 29.27%, 58.54%, and 28.05%, respectively. In a Chinese study [31], resistance to carbapenems was not reported; however, one in nine strains in our study were resistant.

Regarding *Enterobacter* spp., in comparison to our study, a Brazilian study [25] reported higher resistance rates of their isolates to piperacilin/tazobactam (22.45% vs. 16.67%), cefepim (18.36% vs. 11.11%), ciprofloxacin (18.36% vs. 5.56%), and gentamicin (12.24% vs. 5.56%). The resistance rate to imipenem was 6.12%, whereas, in our study, it was not noted.

In a Pakistani study, absolute resistance of AmpC-producing Enterobacterales to ESC and a resistance rate of 96.45% to gentamicin were reported, which were significantly higher in comparison to this study [14]. 

Carbapenemase-producing Enterobacterales were a common finding in our study. OXA-48 was the dominant type, which is in line with the previous reports from Croatia in the past decade [33,34,35,36]. A rapid spread of OXA-48 carbapenemase in Croatia in the past decade was reported, and now it outnumbers KPC and metallo-β-lactamases, which prevailed in the early stage of carbapenemase dissemination at the beginning of the 2010s [37,38]. Ertapenem resistance, observed in some isolates, was not associated with carbapenemase production, but more likely with overproduction of AmpC β-lactamase or ESBL combined with porin loss [12]; however, clarification of the resistance mechanisms was beyond this study. Despite the high rate of ESBLs and carbapenemases, the strains appear to be more susceptible to the majority of antibiotics in comparison to the aforementioned studies. Interestingly, resistant isolates were commonly associated with elderly patients with severe comorbidities; however, few ESBL- and carbapenemase-producing Enterobacterales were observed in the children’s hospital near Zagreb. A reason for such a discrepancy could be the fact that the prevalence of ESBL- and carbapenemase-producing Enterobacterales is not affected by the age of the patient, but rather by a long-term exposure to hospital surroundings, as observed in a Turkish study [39]. All involved patients had the same type of carbapenemase, indicating negligence towards infection control.

Carbapenemase-induced resistance was found in *E. cloacae* (4.60%), *K. aerogenes* (12.50%), *S. marcescens* (2.13%), *M. morganii* (22.22%), *Providencia* spp. (27.27%), and *C. freundii* (11.11%). In the previous studies, *E. cloacae* was the dominant species, harboring VIM metallo-β-lactamase [37,38] and exhibiting hyperproduction of chromosomal AmpC β-lactamase [17]. These findings coincide with the present study, in which four carbapenem-resistant MBL-positive strains were reported. Carbapenem resistance in one strain was not a result of carbapenemase production, but rather of AmpC overproduction and porin loss. Other types of carbapenemases were not found in *E. cloacae*. The proportion of carbapenem-resistant *E. cloacae* in this study was 2.5 times lower than that in a Spanish study (4.60% vs. 11.54%) [40]. In an Egyptian study [41], almost 50% of the strains were carbapenemase producers (49.33%); this proportion was 10 times higher than the one reported in our study. Moreover, the German study [42] confirmed a carbapenemase rate of 75.00%, which exceeded the one reported in all of the aforementioned studies.

*Klebsiella aerogenes* strains were carabapenemase producers in 12.50% of the cases, which was significantly less than the rate observed in the Egyptian study (30.00%) [41].

Only sporadic carbapenamase-producing strains of *S. marcescens* were observed in our study (1/47, 2.13%) whereas the German study [42] noted that 33 out of 45 strains (73.33%) were carbapenemase positive.

One *C. freundii* isolate produced double carbapenemase (KPC + NDM). Such isolates were reported previously with increasing frequency in UHCZ, but in the previous studies a combination of OXA-48 with MBLs prevailed [43]. Carbapenemases were produced by 11.11% of our strains, which was higher than the rate in the Spanish study (4.35%) [40] and notably less in contrast to the German study (86.54%) [42]. Double carbapenemase was also observed in the German study [42].

Management of BSI due to microorganisms harboring AmpC β-lactamases, extended-spectrum β-lactamases, or carbapenemases remains challenging, and standardized guidelines could prove to be a valuable asset. According to the guidelines published by the Infectious Disease Society of America (IDSA) [44], it is mandatory that an antibiotic of choice demonstrates in vitro activity against the identified causative agent. Choosing an empirical therapy is a difficult task, yet it should be undertaken based on three factors: previous organisms identified from the patient supported by antibiotic susceptibility testing, antibiotic exposures within the past 30 days, and local susceptibility patterns [44]. 

Following antibiotic susceptibility testing, the treatments of choice for AmpC β-lactamases include both carbapenems and cefepime [44]. As a result of meropenem’s potency of microbiological eradication, it is considered a first-line therapy. Consequently, it was the most frequently administered antibiotic in our study, both as an empirical and a targeted therapy. Attributable to growing concern regarding carbapenem-resistant Enterobacterales, integrating non-carbapenem treatment strategies is being explored for these pathogens [45,46]. Cefepime is not hydrolyzed by AmpC β-lactamases and has low AmpC induction potential, which makes it an excellent substitution for carbapenems [47]. As claimed by the IDSA, piperacilin/tazobactam is not suggested for treatment of infections caused by AmpC-producing Enterobacterales; however, some reports indicate it possesses activity that is almost equal to that of carbapenems, just like cefepime [45,46]. In our study, achievement of clinical cure and complete pathogen eradication were reported with administration of both cefepime and piperacilin/ tazobactam, which is in line with the aforementioned studies [45,46]. 

Treatment of ESBL-producing Enterobacterales includes ciprofloxacin carbapenems, aminoglycosides, and ceftazidim/avibactam [44]. Previous studies reported an incidence of ESBLs among *Enterobacter* spp. strains of 15.27% [48]. They report that the production of ESBLs was associated with an inappropriate empirical therapy but not with a poorer outcome [48]. In our study, the incidence of ESBLs was low among *M. morganii* (0.00%) and *Enterobacter* spp. (5.56%) strains, but relatively high in *E. cloacae* (11.49%), *K. aerogenes* (18.75%), *Serratia marcescens* (19.15%)*, Providencia* spp. (18.18%), and *C. freundii* (11.11%) strains. ESBL positivity was associated with resistance to cefepime.

Ceftazidim/avibactam is the antibiotic of choice for OXA-48-producing organisms, subsequent to antibiotic susceptibility testing [44,49]. Occurrence of carbapenemases with a frequency ranging from 2.13% to 27.27% among our isolates is daunting, as they compromise the administration of carbapenems and ESC. In our study, ceftazidime/avibactam and colistin were administered in the case of positive carbapenemase according to the OKNV test. Colistin resistance in Croatia was predominantly associated with OXA-48-producing *K. pneumoniae* in the previous studies [50,51].

Antibiotic switch was indicated in 61 cases (35.47%), commonly in patients with carbapenem-resistant or ESBL-positive isolates. 

The mortality rate reported in our study differs from that found in a Korean study (21.51% vs. 16.75%) [48], but is equal to that in the Brazilian study (21.51% vs. 21.50%) [25]. Furthermore, lethal outcome in the American study (35/150, 23.33%) was more frequent than in our study. Additionally, mortality rates of BSI associated with *E. cloacae* and *K. aerogenes* noted in the American study [52] were higher in comparison to those in our study (21.15% vs. 17.24% and 28.26% vs. 18.75%, respectively). No significant difference was observed between mortality rates of infections with resistant and susceptible strains (24.07% vs. 20.34%, *p* = 0.58). The reason for this absence could be the extensive administration of meropenem as an empirical therapy, which exerts bactericidal activity against ESBL positive strains and AmpC hyperproducers [7,9]. Carbapenems are used in our hospital as empirical therapy because ESBL-producing organisms are endemic, particularly in ICUs. However, they exert selection pressure, which favors the spread of carbapenemase-producing organisms. Although no official guidelines exist in Croatia, mortality rates do not differ significantly from those in the rest of the world, due to the accessibility of guidelines published in other countries, such as the United States, which are then used, to an extent, by Croatian clinicians.

This study has several limitations. Firstly, the number of patients enrolled was quite small as a result of the small proportion of bloodstream infections caused by AmpC-producing Enterobacterales, which contributed to less than 1% of total BSI episodes. The chi-square test of independence is less accurate when applied to small sample sizes. In small samples, the distribution of the test statistic might deviate from the theoretical chi-square distribution, affecting the validity of the results. Secondly, inhibitor-based tests with phenylboronic acid or cloxacillin should have been used to affirm AmpC hyperproducing strains; however, the strains were not stored for further analysis. Another limitation is the fact that molecular analysis of the resistance traits was not conducted.

In summary, bloodstream infections caused by AmpC-producing Enterobacterales are becoming quite a challenge for clinicians to handle due to their ever-growing resistance to known antibiotics. Strains’ resistance patterns display geographical differences; therefore, evaluation from time to time is advisable in order to predict the change in susceptibility patterns of bacterial strains. Further research of this matter should be conducted and proactive measures of infection control should be followed to the letter in order to effectively combat the rising threat.

## Figures and Tables

**Figure 1 pathogens-12-01125-f001:**
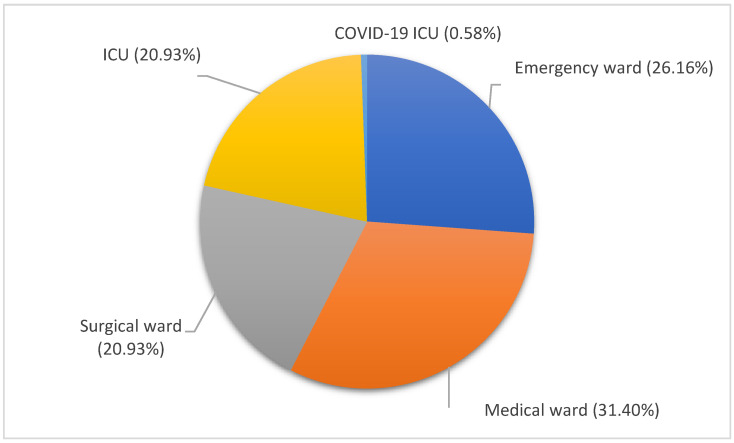
Distribution of BSI episodes according to the hospital wards. Medical ward, orange, 31.40%; emergency ward, dark blue, 26.16%; surgical ward, gray, 20.93%; ICU, yellow, 20.93%; COVID-19 ICU, blue, 0.58%.

**Figure 2 pathogens-12-01125-f002:**
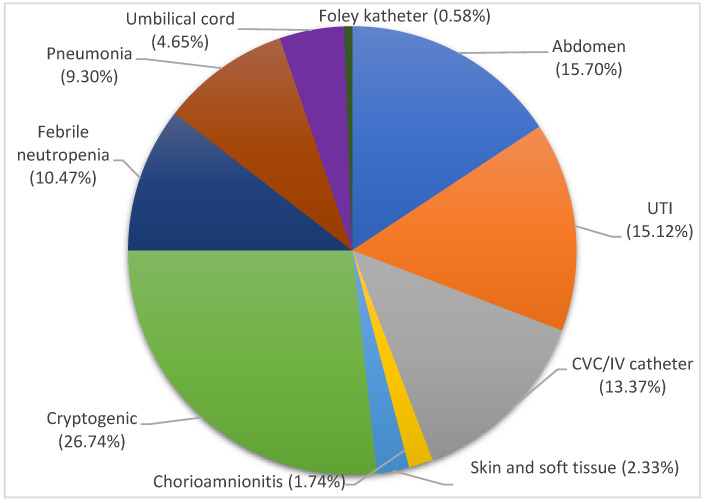
Source of bloodstream infections. Cryptogenic, light green, 26.74%; abdomen, blue, 15.70%; UTI, orange, 15.12%; CVC/IV catheter, gray, 13.37%; febrile neutropenia, dark blue, 10.47%; pneumonia, brown, 9.30%; umbilical cord, purple, 4.65%; skin and soft tissue, light blue, 2.33%; chorioamnionitis, yellow, 1.74%; Foley catheter, dark green, 0.58%.

**Figure 3 pathogens-12-01125-f003:**
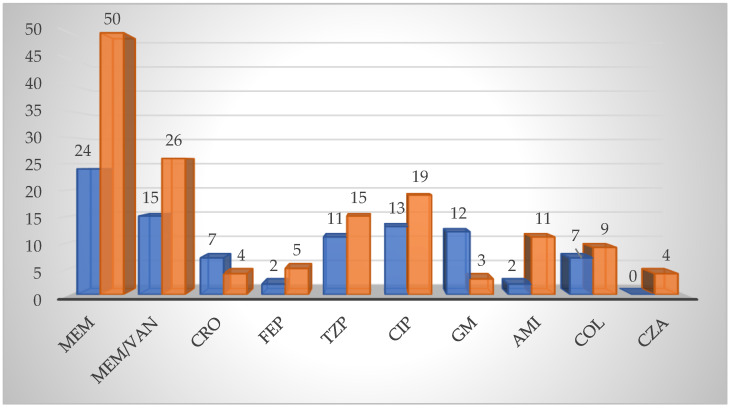
Antibiotic therapy regimes. Empirical antimicrobial therapy, blue; Targeted antimicrobial therapy, orange. Abbreviations: MEM—meropenem; VAN—vancomycin; CRO—ceftriaxone; FEP—cefepime; TZP—piperacilin/tazobactam; CIP—ciprofloxacin; GM—gentamicin; AMI—amikacin; COL—colistin; CZA—ceftazidime/avibactam.

**Table 1 pathogens-12-01125-t001:** Antibiotic susceptibility and β-lactamase production of AmpC-producing Enterobacterales.

	TZP	CAZ	CTX	CRO	FEP	ETP	IMI	MEM	GM	AMI	CIP	COL	ESBL	AmpC Hyper-Production	Carbapenemase	Type of Carbapenemase
*Enterobacter cloacae* n = 87	24.14% (21/87)	34.48% (30/87)	35.63% (31/87)	34.48% (30/87)	27.59% (24/87)	21.84% (19/87)	5.75% (5/87)	5.75% (5/87)	20.69% (18/87)	1.15% (1/87)	20.69% (18/87)	0.00% (0/87)	11.49% (10/87)	22.99% (20/87)	4.60% (4/87)	VIM (n = 2) NDM (n = 2)
*Klebsiella aerogenes* n = 16	62.50% (10/16)	62.50% (10/16)	62.50% (10/16)	62.50% (10/16)	31.25% (5/16)	43.75% (7/16)	6.25% (1/16)	6.25% (1/16)	25.00% (4/16)	12.50% (2/16)	18.75% (3/16)	0.00% (0/16)	18.75% (3/16)	31.25% (5/16)	12.50% (2/16)	OXA-48 (n = 2)
*Serratia marcescens* n = 47	8.51% (4/47)	27.66% (13/47)	25.53% (12/47)	27.66% (13/47)	23.40% (11/47)	6.38% (3/47)	4.25% (2/47)	4.25% (2/47)	25.53% (12/47)	14.89% (7/47)	8.51% (4/47)	0.00% (0/47)	19.15% (9/47)	17.02% (8/47)	2.13% (1/47)	OXA-48 (n = 1)
*Morganella morganii* n = 9	22.22% (2/9)	11.11% (1/9)	11.11% (1/9)	11.11% (1/9)	22.22% (2/9)	22.22% (2/9)	0.00% (0/9)	0.00% (0/9)	0.00% (0/9)	0.00% (0/9)	0.00% (0/9)	0.00% (0/9)	0.00% (0/9)	0.00% (0/9)	22.22% (2/9)	OXA-48 (n = 2)
*Providencia* spp. n = 11	27.27% (3/11)	63.64% (7/11)	63.64% (7/11)	63.64% (7/11)	36.36% (4/11)	27.27% (3/11)	27.27% (3/11)	0.00% (0/11)	63.64% (7/11)	27.27% (3/11)	72.73% (8/11)	0.00% (0/11)	18.18% (2/11)	45.45% (5/11)	27.27% (3/11)	OXA-48 (n = 3)
*Citrobacter freundii* n = 9	33.33% (3/9)	33.33% (3/9)	33.33% (3/9)	33.33% (3/9)	22.22% (2/9)	11.11% (1/9)	11.11% (1/9)	11.11% (1/9)	33.33% (3/9)	11.11% (1/9)	33.33% (3/9)	0.00% (0/9)	11.11% (1/9)	22.22% (2/9)	11.11% (1/9)	KPC (n = 1) NDM (n = 1)
*Enterobacter* spp. n = 18	16.67% (3/18)	16.67% (3/18)	16.67% (3/18)	16.67% (3/18)	11.11% (2/18)	11.11% (2/18)	0.00% (0/18)	0.00% (0/18)	5.56% (1/18)	0.00% (0/18)	5.56% (1/18)	0.00% (0/18)	5.56% (1/18)	16.67% (3/18)	0.00% (0/18)	

Abbreviations: TZP—piperacillin/tazobactam; CAZ—ceftazidime; CTX—cefotaxime; CRO—ceftriaxone; FEP—cefepime; IMI—imipenem; MEM—meropenem; GM—gentamicin; AMI—amikacin; CIP—ciprofloxacin; COL—colistin; ESBL-double-disk synergy test for detection of extended-spectrum β-lactamases; AmpC—Amp-C hyperproduction; VIM—Verona integron-encoded metallo-β-lactmase; NDM—New Delhi metallo-β-lactmase; OXA-48—oxacillinase-48; KPC—*Klebsiella pneumoniae* carbapenemase.

## Data Availability

The data presented in this study are available on request from the corresponding author.

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
