# Peer review of "Bloodstream Infections by AmpC-Producing Enterobacterales: Risk Factors and Therapeutic Outcome"

_pathogens, 2023, doi:10.3390/pathogens12091125_

Round 1

Reviewer 1 Report

The work is clear. It describes the situation of Enterobacterales strains in Croatia in patients with BSI. Detection of bacterial isolates and determination of antibiotic sensitivity is methodologically correct. The discussion is  qualitatively processed.

Author Response

REPLY TO REVIEWER 1

Dear Madam/Sir

The work is clear. It describes the situation of Enterobacterales strains in Croatia in patients with BSI. Detection of bacterial isolates and determination of antibiotic sensitivity is methodologically correct. The discussion is  qualitatively processed.

Thank you for your valuable opinion.

Reviewer 2 Report

The manuscript by Pospisil at al addresses a topic of interest, as limited data are available in the literature to date. However, the manuscript has gaps and should be improved with a major revision.

Main points:

1. Extensive editing of English language is required.

2. Title: “Bloodstrem infections by” is most appropriate of “Bloodstrem infections with”. Please revise.

3. The study analyses bloodstream infections by Chromosomal AmpCS –producing Enterobacterales. Please revise the Introduction extending description of Chromosomal AmpCs (main enzymes and spectrum of hydrolytic activity)

4. The incidence of chromosomal AmpCS-producing Enterobacterales must be analysed, using total blood cultures, positive for Enterobacteriaceae, or Gram-negative species as the denominator.

5. The study analyses bloodstream infections by Chromosomal AmpCS –producing Enterobacterales. Species involved in this group are Enterobacter cloacae, K. aerogenes, Providencia spp., Serratia marcescens, Citrobacter freundii, Morganella morganii as also stated in lines 57-60. Other species not included in this list should be excluded. In the study, other species such as Enterobacter spp, Serratia spp, Citrobacter spp, were included. Please revise.

6. Lines 129: Ceftazidime/avibactam, ceftolozane/tazobactam were tested in cases of carbapenemase-producers. However, ceftolozane/tazobactam is not active and not recommended for carbapenemase-producers. Please clarify.

Ceftolozane/tazobactam is not appropriate for carbapenemase-producers.

6. Modified Hodge test is not recommended by EUCAST or CLSI as carbapenemase detection method as it has low sensitivity and specificity. If there were no discordance with OKNV test, the use of this method may be removed from the manuscript.

7. When carbapenemase detection method was performed? Please add specific criteria (EUCAST or CLSI). “reduced inhibition zones around carbapenem disk” (lines 136) is not a specific criteria.

7. In cases of carbapenemase or ESBL production the overproduction of AmpC cannot be confirmed using the methods described in lines 132-136. Please discuss this limitation in the discussion.

8. The discussion should be enriched by comparing antibiotic sensitivity data, and the incidence of carbapenemases, with other studies, at least European ones.

Minor comments:

Figure 1: Remove the title inside the picture as it is already present in the caption below.

Lines 59: Replace or with and

Lines 61-62: AmpC does not confer resistance to all β-lactam/βlactamase inhibitor combinations. Please revise the sentence.

Line 62: Replace beta with β

Lines 72-74: Revise the sentence by indicating the Ambler classes, and the main types of carbapenemases in brackets for each class.

Line 105: BACTEC FX is not made by Biomerieux. Please clarify.

Line 107: Revise sentence: Blood agar, agar chocolate. Is Columbia Agar blood agar? Please clarify.

Lines 115-120: This part repeats content already present in the previous paragraph (Lines 104-109). Review by moving lines 104-109 to the paragraph "Bacterial isolates".

Lines 185: Add results after susceptibility testing

Table 1. Indicate the number of isolates here (last column) and also in the manuscript in the following form: (n= ...).

Lines 294: Replace Enterobacteriaceae with Enterobacterales.

Extensive editing of English language is required.

Author Response

Dear Madam/Sir

Thank you for your valuable comments

In the manuscript by Pospisil at al addresses a topic of interest, as limited data are available in the literature to date. However, the manuscript has gaps and should be improved with a major revision.

Main points:

1.Extensive editing of English language is required.

  1. Grammatical and spelling errors have been corrected
  2. Title: “Bloodstrem infections by” is most appropriate of “Bloodstrem infections with”. Please revise.
  3. A. corrected: by instead of with
  4. The study analyses bloodstream infections by Chromosomal AmpCS –producing Enterobacterales. Please revise the Introduction extending description of Chromosomal AmpCs (main enzymes and spectrum of hydrolytic activity)
  5. The text is added:

Their hydrolytic activity encompasses penicillins, first, second and third generation cephalosporins and cephamycins but spare cefepime and carbapenems. Unlike ESBLs they are not inhibited by clavulanic acid, sulbactam or tazobactam. Plasmid-mediated AmpC β-lactamases are derived from the chromosomal β-lactamases of the bacteria belonging to the genus Enterobacter, Serratia, Citrobacter, Pseudomonas and Acinetobacter by the escape of the chromosomal gene to the plasmid. The most prevalent types are DHA, ACT, FOX, MOX and CMY, usually detected in K. pneumoniae and P. mirabilis.  

  1. The incidence of chromosomal AmpCS-producing Enterobacterales must be analysed, using total blood cultures, positive for Enterobacteriaceae, or Gram-negative species as the denominator.
  2. The incidence of chromosomal AmpC-producing Enterobacterales against total number of blood cultures was calculated. The proportion of blood cultures with AmpC producing Enterobacterales was 197/39098 (0.50%) where the denominator is the total number of BCs done in certain timeframe in UHCZ.

  1. The study analyses bloodstream infections by Chromosomal AmpCS –producing Enterobacterales. Species involved in this group are Enterobacter cloacae, K. aerogenes, Providencia spp., Serratia marcescens, Citrobacter freundii, Morganella morganii as also stated in lines 57-60. Other species not included in this list should be excluded. In the study, other species such as Enterobacter spp, Serratia spp, Citrobacter spp, were included. Please revise.
  2. We have corrected: Serratia spp, Citrobacter spp, Morganella spp although all Serratia strains were species S. marcescens and all Citrobacter strains were freundii according to MALDI-TOF.
  3. Lines 129: Ceftazidime/avibactam, ceftolozane/tazobactam were tested in cases of carbapenemase-producers. However, ceftolozane/tazobactam is not active and not recommended for carbapenemase-producers. Please clarify. Ceftolozane/tazobactam is not appropriate for carbapenemase-producers.
  4. We have removed ceftolozane/tazobactam. It was tested because it has activity on the carbapenem resistant strains due to porin loss or upregulation of efflux pumps.
  5. Modified Hodge test is not recommended by EUCAST or CLSI as carbapenemase detection method as it has low sensitivity and specificity. If there were no discordance with OKNV test, the use of this method may be removed from the manuscript.
  6. We have removed modified Hodge test due to its low sensitivity and specificity.
  7. When carbapenemase detection method was performed? Please add specific criteria (EUCAST or CLSI). “reduced inhibition zones around carbapenem disk” (lines 136) is not a specific criteria.
  8. We have used EUCAST criteria (< 22 mm for imipenem and meropenem and < 25 mm for ertapenem)
  9. In cases of carbapenemase or ESBL production the overproduction of AmpC cannot be confirmed using the methods described in lines 132-136. Please discuss this limitation in the discussion.
  10. A. Suspected AmpC hyperproducing strains should be confirmed by using inhibitor- based tests with phenylboronic acid or cloxacillin but unfortunately the strains were not stored for further analysis
  11. The discussion should be enriched by comparing antibiotic sensitivity data, and the incidence of carbapenemases, with other studies, at least European ones.
  12. The results were compared with other studies but we have not found any references from Europe in PubMed.

Minor comments:

Figure 1: Remove the title inside the picture as it is already present in the caption below.

A.The title has been removed.

Lines 59: Replace or with and

A.corrected

Lines 61-62: AmpC does not confer resistance to all β-lactam/βlactamase inhibitor combinations. Please revise the sentence.

A.We have explained that some β-lactam combinations with inhibitors may be active against AmpC producing organisms.

Line 62: Replace beta with β

A.replaced: β instead of beta

Lines 72-74: Revise the sentence by indicating the Ambler classes, and the main types of carbapenemases in brackets for each class.

A.The main types of carbapenemase are already mentioned.

Carbapenemases in Enterobacterales belong to Ambler class A serine β-lactamases (KPC, GES, SME, IMI, NMC), class B metallo-β-lactamases (MBL) of the IMP, VIM or NDM family or OXA-48-like β-lactamases belonging to the class D or carbapenem-hydrolyzing oxacillinases.

Line 105: BACTEC FX is not made by Biomerieux. Please clarify.

A.That was error. It is Becton Dickinson.

Line 107: Revise sentence: Blood agar, agar chocolate. Is Columbia Agar blood agar? Please clarify.

A.corrected: blood agar, chocolate agar and Columbia anaerobic blood agar

Lines 115-120: This part repeats content already present in the previous paragraph (Lines 104-109). Review by moving lines 104-109 to the paragraph "Bacterial isolates".

A.The repetitive text has been removed from the lines 115-120.

Lines 185: Add results after susceptibility testing

A.antibiotic therapy was moved after susceptibility testing.

Table 1. Indicate the number of isolates here (last column) and also in the manuscript in the following form: (n= ...).

A.Corrected, number of isolates with carbapenemases is presented as n=

Lines 294: Replace Enterobacteriaceae with Enterobacterales.

A.corrected

Reviewer 3 Report

In the manuscript entitled "Bloodstream infections with AmpC producing Enterobacterales: risk factors and therapeutic outcome", the authors present a retrospective study of patients in Croatia with bloodstream infections. The patients and bacterial cause of their infections were described. The topic is important to study, but the authors have not yet crafted a manuscript with statistically meaningful, publication-ready data. Some items that must be addressed are as follows:

1.     Statistical analysis is lacking or without description.

a.     Several p-values are given in lines 163-174, but the method of their calculation is not described anywhere in the manuscript.

b.     Without statistical significance, all that the authors have presented is a collection of observations. A thorough statistical analysis needs to be performed so that conclusions can be drawn.

c.      Limitations of the study are not fully or clearly stated – a large part of these limitations is the current lack of statistical analysis.

2.     Care must be taken to distinguish what information is part of the retrospective study and what (if any) is original work by the authors. The current text is confusing in this regard.

3.     The work is incompletely and/or inconsistently presented.

a.     Citations are lacking in the introduction and methods sections.

b.     Figure legends are lacking descriptive text.

c.      The number of patients used for each set of information varies between 172 and 197. The data would be more meaningful if the dataset were consistent across all analyses.

d.     The number of decimal places should be consistent. In some places the percent values are rounded to the nearest whole number and in other places the percent values are rounded to the nearest tenth. (e.g., Figure 1 and the surrounding text)

e.     In line 53, the term ESKAPE pathogens is used improperly. The acronym has a K not a C.

4.     Substantial formatting and English editing must be performed.

a.     The Introduction should end with a summary of the findings. As noted previously, a proper analysis of the data is missing from the manuscript entirely, and will need to be performed before such a summary can be crafted.

b.     Patient selection (the first Results subheading) is more applicably placed in the Methods section.

c.      In the Results and Discussions sections, readers would be more able to follow the work if the authors were to include their reasoning for performing each test and more cohesively group their experiments.

d.     The tables should be rotated to fit onto single pages. In Landscape orientation, they will be more legible. This correction can be made later in final edits…

e.     Formal English writing should be used in scientific publications. For example, lines 139-140, while descriptive, are quite colloquial. Also, phrases such as there is/was and it is/was are to be avoided in formal written English.

The text is understandable, English editing needs only to be performed do to the fact that the text is too informal. Formal English grammar and syntax should be used in scientific writing/publications.

Author Response

Dear Madams/Sirs,

Thank you for your valuable comments

In the manuscript entitled "Bloodstream infections by AmpC producing Enterobacterales: risk factors and therapeutic outcome", the authors present a study of patients in Croatia with bloodstream infections. Both the patients and bacterial isolates, a cause of their infections, were described.

Our manuscript was corrected accordingly to the comments made by reviewer No. 2. All problems were adressed (red).

  1. Statistical analysis was described lacking or without description.
  2. statistical analysis has been described

2.Several p-values are given in lines 163-174, but the method of their calculation is not described anywhere in the manuscript.

  1. The method of calculation is added.

3.Without statistical significance, all that the authors have presented is a collection of observations. A thorough statistical analysis needs to be performed so that conclusions can be drawn.

A.The problem with statistical analysis is the small number of patients and samples and this limits the significance of the differences between groups of patients.

  1. Limitations of the study are not fully or clearly stated – a large part of these limitations is the current lack of statistical analysis.
  2. statistical analysis has been improved and limitations are explained
  3. Care must be taken to distinguish what information is part of the retrospective study and what (if any) is original work by the authors. The current text is confusing in this regard.

A.We have explained that this is retrospective study and that all data regarding the strains were retrieved from the hospital internet system. The strains were not available to perform more detailed laboratory analysis for detection of resistance genes because they were not stored.

6.The work is incompletely and/or inconsistently presented.

A.We have tried to improve presentation.

7.Citations are lacking in the introduction and methods sections.

  1. More citations are added in the introduction section.
  2. Figure legends are lacking descriptive text.

Description of figures is given in captations at the bottom of the figures

9.The number of patients used for each set of information varies between 172 and 197. The data would be more meaningful if the dataset were consistent across all analyses.

  1. Unfortunately for 25 patients who provided blood cultures we do not have clinical data because they were hospitalized in the special hospital for chronically ill children. The hospital does not have a microbiology laboratory and they send the blood samples to University Hospital Centre Zagreb. We cannot get the clinical data from the other hospitals unless we get the permission from their Ethical Comittee. That would last too long.
  2. The number of decimal places should be consistent. In some places the percent values are rounded to the nearest whole number and in other places the percent values are rounded to the nearest tenth. (e.g., Figure 1 and the surrounding text)
  3. We have corrected the percent values to two decimal place.

11.In line 53, the term ESKAPE pathogens is used improperly. The acronym has a K not a C.

  1. corrected: ESKAPE
  2. Substantial formatting and English editing must be performed.

A.English language has been corrected to some extent

13.The Introduction should end with a summary of the findings. As noted previously, a proper analysis of the data is missing from the manuscript entirely, and will need to be performed before such a summary can be crafted.

  1. The summary of the findings has been added to the abstract. We have added the reasons to carry out the study at the end of the introduction section.

Round 2

Reviewer 2 Report

The authors addressed most of my issues.

Author Response

REPLY TO REVIEWER 2

The authors addressed most of my issues.

Dear Madam/Sir,

Thank you for your valuable opinion.

Reviewer 3 Report

In this revision of the manuscript entitled "Bloodstream infections by AmpC producing Enterobacterales: risk factors and therapeutic outcome", the authors have made substantial improvements over the previous version of the manuscript, including adding figure legends, adding information in the introduction, and critically evaluating their data in the discussion. These improvements greatly improve the quality of the manuscript.

Yet several problems remain and should be addressed prior to publication:

1. To begin, I commend the authors on their thoughtful additions to the introduction and discussion sections of the manuscript. These additions have most definitely strengthened the material presented.  Still, two issues remain.

a. Citations are incomplete in places. As the document stands, credit is not properly given to past work by other groups.

b. Several parts of the additional text read as simple summaries of facts. The authors are encouraged to consider summarizing and organizing the information in a more concise and palatable manner for their readers.

2. The methods section still has room to be improved.

a. The chi-squared test is now described in the methods section in this revision.  This description, however, is a basic description of the theory and not how it was used in the authors’ hands. Description of how tools are used by the authors allows greater repeatability of results.

b. Lines 167-169 describe a limit of the study, not a method performed.

c. Definition of the qualitative resistance levels on lines 232 and 239 should be included in the methods section.

3. The results section is thorough, but several minor modifications should be addressed prior to publication.

a. Figure legends should be descriptive.  This change can be accomplished by a simple reformatting of the text.  For the pie charts, the correct descriptive text would be something along the lines of <category>, <wedge color>, <% value>; For example, with Figure 1: Medical ward, orange, 31.40% patients; Emergency ward, dark blue, 26.15% patients; etc. Analysis/Qualification of results should not be included in the figure legends.

b. Abbreviations from the additional column in Table 1 have not been included in the list at the bottom of the table.

c. The authors may wish to consider presenting the data in the table to emphasize the standardized results (percentages) in the top half of the data rows and the raw data (fractions) in the bottom half of the data rows.  This change is neither significant nor necessary, the data may be easier to interpret in this orientation.

d. The section on antibiotic resistance data needs to be crafted so as to indicate that the authors are using data from patient records. The current syntax is very active (i.e., indicative of work being performed) rather than passive (i.e., indicative of data gathered).

e. The description of antibiotic therapy regimens is very complex and confusing.  Reader understanding could be augmented with one or more flowcharts/schematics of the regimens, which can be included as supplemental information.

f. What is the standard of care for these infections?  Including the expected care (or noting the lack of standardization) would improve the analysis of the observed results.

Again, significant improvements have been made to the initial manuscript, and the work will be worthy of publication after minor revisions.

Quality of English language is sufficient for publication.  This reviewer has no concerns about clarity or syntax.

Author Response

Dear Madam/Sir,

Thank you for your valuable comments.

Corrections are blue.

  1. To begin, I commend the authors on their thoughtful additions to the introduction and discussion sections of the manuscript. These additions have most definitely strengthened the material presented.  Still, two issues remain.
  2. Citations are incomplete in places. As the document stands, credit is not properly given to past work by other groups.

Several citations were added in the introduction and discussion the rest were revised.

  1. Several parts of the additional text read as simple summaries of facts. The authors are encouraged to consider summarizing and organizing the information in a more concise and palatable manner for their readers.

Minor changes were added, however, the comparison was a request by reviewer No. 2. and was one of the reason for the approval of the manuscript.

  1. The methods section still has room to be improved.
  2. The chi-squared test is now described in the methods section in this revision.  This description, however, is a basic description of the theory and not how it was used in the authors’ hands. Description of how tools are used by the authors allows greater repeatability of results.

Added

  1. Lines 167-169 describe a limit of the study, not a method performed.

Added to the limitations, removed from methods.

  1. Definition of the qualitative resistance levels on lines 232 and 239 should be included in the methods section.

Lines 232 and 239 revised.

  1. The results section is thorough, but several minor modifications should be addressed prior to publication.
  2. Figure legends should be descriptive.  This change can be accomplished by a simple reformatting of the text.  For the pie charts, the correct descriptive text would be something along the lines of <category>, <wedge color>, <% value>; For example, with Figure 1: Medical ward, orange, 31.40% patients; Emergency ward, dark blue, 26.15% patients; etc. Analysis/Qualification of results should not be included in the figure legends.

Added

  1. Abbreviations from the additional column in Table 1 have not been included in the list at the bottom of the table.

Added

  1. The authors may wish to consider presenting the data in the table to emphasize the standardized results (percentages) in the top half of the data rows and the raw data (fractions) in the bottom half of the data rows.  This change is neither significant nor necessary, the data may be easier to interpret in this orientation.

Corrected

  1. The section on antibiotic resistance data needs to be crafted so as to indicate that the authors are using data from patient records. The current syntax is very active (i.e., indicative of work being performed) rather than passive (i.e., indicative of data gathered).

Corrected

  1. The description of antibiotic therapy regimens is very complex and confusing.  Reader understanding could be augmented with one or more flowcharts/schematics of the regimens, which can be included as supplemental information.

Figure 3. added

  1. What is the standard of care for these infections?  Including the expected care (or noting the lack of standardization) would improve the analysis of the observed results.

Added in discussion.